# Jumpout : Improved Dropout for Deep Neural Networks with Rectified Linear Units

## Abstract

Dropout is a simple yet effective technique to improve the generalization performance and prevent overfitting in deep neural networks (DNNs). In this paper, we discuss three novel observations about dropout to better understand the generalization of DNNs with rectified linear unit (ReLU) activations: 1) dropout is a smoothing technique that encourages each local linear model of a DNN to be trained on data points from nearby regions; 2) a constant dropout rate can result in effective neural-deactivation rates that are significantly different for layers with different fractions of activated neurons ; and 3) the rescaling factor of dropout causes an inconsistency to occur between the normalization during training and testing conditions when batch normalization is also used. The above leads to three simple but nontrivial improvements to dropout resulting in our proposed method "Jumpout." Jumpout samples the dropout rate using a monotone decreasing distribution (such as the right part of a truncated Gaussian), so the local linear model at each data point is trained, with high probability, to work better for data points from nearby than from more distant regions. Instead of tuning a dropout rate for each layer and applying it to all samples, jumpout moreover adaptively normalizes the dropout rate at each layer and every training sample/batch, so the effective dropout rate applied to the activated neurons are kept the same. Moreover, we rescale the outputs of jumpout for a better trade-off that keeps both the variance and mean of neurons more consistent between training and test phases, which mitigates the incompatibility between dropout and batch normalization. Compared to the original dropout, jumpout shows significantly improved performance on CIFAR10, CIFAR100, Fashion-MNIST, STL10, SVHN, ImageNet-1k, etc., while introducing negligible additional memory and computation costs.

## 1 Introduction

Deep learning has achieved remarkable success on a variety of machine learning tasks (Russakovsky et al., 2015; Rajpurkar et al., 2016). Deep neural networks (DNN), however, are often able to fit the training data perfectly — this can result in the overfitting problem, thereby weakening the generalization performance on unseen data. Dropout (Srivastava et al., 2014; Huang et al., 2016) is a simple yet effective technique to mitigate such problems by randomly setting the activations of hidden neurons to $0$, a strategy that reduces co-adaptation amongst neurons. Dropout applies to any layer in a DNN without causing significant additional computational overhead.

Dropout, however, has several drawbacks. Firstly, dropout rates, constituting extra hyper-parameters at each layer, need to be tuned to get optimal performance. Too high a dropout rate can slow the convergence rate of the model, and often hurt final performance. Too low a rate yields few or no improvements on generalization performance. Ideally, dropout rates should be tuned separately for each layer and also during various training stages. In practice, to reduce computation, we often tune a single dropout rate and keep it constant for all dropout layers and throughout the training process.

If we treat dropout as a type of perturbation on each training sample, it acts to generalize the DNN to noisy samples having that specific expected amount of perturbation (due to the fixed dropout rate) with high probability. The fixed rate rules out samples typical having less perturbation, i.e., those potentially more likely to be closer to the original samples and thus that are potentially more helpful to improve generalization. Also, when a constant dropout rate is applied to layers and samples having

different fractions of activated neurons, the effective dropout rate (i.e., the proportion of the activated neurons that are deactivated by dropout) varies, which might result in too much perturbation for some layers and samples and too little perturbation for others.

Another deficiency of dropout lies in its incompatibility with batch normalization (BN) (Ioffe & Szegedy, 2015) (more empirical evidence of this is shown in Section 3.3). As dropout randomly shuts down activated neurons, it needs to rescale the undropped neurons to match the original overall activation gain of the layer. Unfortunately, such rescaling breaks the consistency of the normalization parameters required between training and test phases[1] and may cause poor behavior when used with BN. Since BN, and its variants (Ba et al., 2016; Ulyanov et al., 2016; Wu & He, 2018), has become an almost indispensable component of modern DNN architectures to keep the training stable and to accelerate convergence, dropout itself often gets dropped out in the choice between these two non-complementary options and has recently become less popular.

## 1.1 OUR APPROACH

We propose three simple modifications to dropout in order to overcome the drawbacks mentioned above. These modifications lead to an improved version of dropout we call "jumpout." Our approach is motivated by three observations about how dropout results in improved generalization performance for DNNs with rectified linear unit (ReLU) activations, which covers a frequently used class of DNNs.

Firstly, we note that any DNN with ReLU is a piecewise linear function which applies different linear models to data points from different polyhedra defined by the ReLU activation patterns. Based on this observation, applying dropout to a training sample randomly changes its ReLU activation patterns and hence the underlying polyhedral structure and corresponding linear models. This means that each linear model is trained not only to produce correct predictions for data points in its associated polyhedron, but also is trained to work for data points in nearby polyhedra; what precisely "nearby" means depends on the dropout rate used. This partially explains why dropout improves generalization performance. The problem, however, is that with a fixed dropout rate, say $p$, and on a layer with $n$ units, the typical number of units dropped out is $np$ as that is the mode of a Binomial distribution with parameter $p$. It is relatively rare that either very few (closer to zero) or very many (closer to $n$) units are dropped out. Thus, with high probability, each linear model is smoothed to work on data points in polyhedra at a typical distance away. The probability of smoothing over closer distances is potentially much smaller, thus not achieving the goal of local smoothness.

In jumpout, by contrast, $p$ rather than being fixed is itself a random variable; we sample $p$ from a distribution that is monotone decreasing (e.g., a truncated half-Gaussian). This achieves the property that $\Pr(i \text{ units dropping out}) \geq \Pr(i+1 \text{ units dropping out})$ for all $i \in \{1, 2, \ldots, n\}$. That is, a smaller dropout rate has a higher probability of being chosen. Hence, the probability of smoothing polyhedra to other points decreases as the points move farther away.

Secondly, we notice that in dropout, the fraction of activated neurons in different layers, for different samples and different training stages, can be different. Although we are using the same dropout rate, since dropping out neurons that are already quiescent by ReLU changes nothing, the effective dropout rate, i.e., the fraction of the activated neurons that are dropped, can vary significantly. In jumpout, we adaptively normalize the dropout rate for each layer and each training sample/batch, so the effective neural-deactivation rate applied to the activated neurons are consistent over different layers and different samples as training proceeds.

Lastly, we address the incompatibility problem between dropout and BN by rescaling the outputs of jumpout in order to keep the variance unchanged after the process of neural deactivation. Therefore, the BN layers learned in the training phase can be directly applied in the test phase without an inconsistency, and we can reap the benefits of both dropout and BN when training a DNN.

In our implementation, similar to dropout, jumpout also randomly generates a 0/1 mask over the hidden neurons to drop activations. It does not require any extra training, can be easily implemented and incorporated into existing architectures with only a minor modification to dropout code. In our experiments on a broad range of benchmark datasets including CIFAR10, CIFAR100, Fashion-

---

[1]Dropout usually happens only during the training but not the test phase, since using it for testing requires averaging the results of multiple dropout inferences on each training sample, which is costly and may introduce greater prediction variance.

MNIST, SVHN, STL10 and ImageNet-1k, jumpout shows almost the same memory and computation costs as the original dropout, but significantly and consistently outperforms dropout on a variety of tasks, as we show below.

## 1.2 RELATED WORK

Jumpout is not the first approach to address the fixed dropout rate problem. Indeed, recent work has proposed different methods to generate adaptive dropout rates. Ba & Frey (2013) proposed "standout" to adaptively change the dropout rates for various layers and training stages. They utilized a binary belief network and trained it together with the original network to control the dropout rates. Zhuo et al. (2015) further extend the model so that adaptive dropout rates can be learned for different neurons or group of neurons. Zhai & Wang (2018) showed that the Rademacher complexity of a DNN is bounded by a function related to the dropout rate vectors, and they proposed to adaptively change dropout rates according to the Rademacher complexity of the network. In contrast to the above methods, jumpout does not rely on additional trained models: it adjusts the dropout rate solely based on the ReLU activation patterns. Moreover, jumpout introduces negligible computation and memory overhead relative to the original dropout methods, and can be easily incorporated into existing model architectures.

Wang & Manning (2013) showed that dropout has a Gaussian approximation called Gaussian dropout and proposed to optimize the Gaussian dropout directly to achieve faster convergence. The Gaussian dropout was also extended and studied from the perspective of variational methods. Kingma et al. (2015) generalized Gaussian dropout and proposed variational dropout, where they connected the global uncertainty with the dropout rates so that dropout rates can be adaptive for every neuron. Molchanov et al. (2017) further extended variational dropout to reduce the variance of the gradient estimator and achieved sparse dropout rates. Other recent variants of dropout include Swapout (Singh et al., 2016), which combines dropout with random skipping connections to generalize to different neural network architectures, and Fraternal Dropout (Zolna et al., 2018), which trains two identical DNNs using different dropout masks to produce the same outputs and tries to shrink the gap between the training and test phases of dropout.

In this paper, we focus on changes to the original dropout that do not require any extra training/optimization costs or introduce more parameters to learn. Jumpout involves orthogonal and synergistic contributions to most of the above methods, and targets different problems of dropout. Indeed, jumpout can be applied along with most other previous variants of dropout.

## 2 ReLU DNNs ARE COMPRISED OF LOCAL LINEAR MODELS

We study a feed-forward deep neural networks of the form:

$$\hat{y}(x) = W_m \psi_{m-1}(W_{m-1} \psi_{m-2}(\ldots \psi_1(W_1 x))), \qquad (1)$$

where $W_j$ is the weight matrix for layer $j$, $\psi_j$ is the corresponding activation function (ReLU in this paper), $x \in X$ is an input data point of $d_{in}$ dimensions and $\hat{y}(x)$ is the network's output prediction of $d_{out}$ dimensions, e.g., the logits before applying softmax. We denote the hidden nodes on layer $j$ to be $h_j$, i.e., $h_j = W_j \psi_{j-1}(W_{j-1} \psi_{j-2}(\ldots \psi_1(W_1 x)))$, whose dimensionality is $d_j$; they represent the nodes after applying the activation function as $\bar{h}_j = \psi(h_j)$.

The above DNN formalization can generalize many DNN architectures used in practice. Clearly, Eqn. (1) can represent a fully-connected network of $m$ layers. Note Eqn. (1) covers the DNNs with bias terms at each layer since the bias terms can be written in the matrix multiplication as well by introducing dummy dimensions on the input data (append $m$ 1's to input data). Moreover, the convolution operator is essentially a matrix multiplication, where every row of the matrix corresponds to applying a convolutional filter on a certain part of the input, and therefore the resulting weight matrix is very sparse and has tied parameters, and typically has an enormous (compared to input size) number of rows. The average-pooling is a linear operator and therefore representable as a matrix multiplication, and max-pooling can be treated as an activation function. Finally, we can represent the residual network block by appending an identity matrix at the bottom of a weight matrix so that we can retain the input values, and add the retained input values later through another matrix operation. Therefore, we can also write a DNN with short-cut connections in the form of Eqn. (1).

For piecewise linear activation functions such as ReLU, the DNN in Eqn. (1) can be written as a piecewise linear function, i.e., the DNN in a region surrounding a given data point $x$ is a linear model

having the following form:

$$\hat{y}(x) = W_m W_{m-1}^x \dots W_1^x x = \frac{\partial \hat{y}(x)}{\partial x} x, \qquad (2)$$

where $W_j^x$ is the equivalent weight matrix after combining the resultant activation pattern with $W_j$. For instance, suppose we use ReLU activation $\psi_{ReLU}(z) = \max(0, z)$; at every layer, we have an activation pattern for the input $a_j(x) \in \{0, 1\}^{d_j}$, and $a_j(x)[p] = 0$ indicates that ReLU sets the unit $p$ to 0 or otherwise preserves the unit value. Then, $\psi_{ReLU}(W_1 x) = W_1^x x$, where $W_1^x$ is modified from $W_1$ by setting the rows, whose corresponding activation patterns are 0, to be all zero vectors. We can continue such a process to the deeper layers, and in the end we can eliminate all the ReLU functions and produce a linear model as shown in Eqn. (2).

In addition, the gradient $\frac{\partial \hat{y}(x)}{\partial x}$ is the weight vector of the linear model. Note that the linear model in Eqn. 2 is specifically associated with the activation patterns $\{a_j(x)\}_{j=1}^m$ on all layers for a data input $x$, which is equal to a set of linear constraints that defines a convex polyhedron containing $x$. In a DNN with ReLU activations, for every dimension $i$ in layer 1, if $a_1(x)[i] = 1$, we have a linear equation $W_1[i]x > 0$ and otherwise we have $W_1[i]x \leq 0$. As a result, we have $d_1$ linear constraints for layer 1. Similarly, we can follow the same procedure on layer $j$ with input changed to $W_{j-1}^x \dots W_1^x x$, so we have $d_j$ linear constraints for layer $j$. Therefore, a DNN with piecewise linear activation functions is a piecewise linear function defined by a number of local linear models (on a set of input data points) and the corresponding convex polyhedra, each represented by a set of linear constraints ($\sum_{j=1}^m d_j$ constraints in specific). An analysis based on a similar perspective can be found in Raghu et al. (2017).

Although the above analysis can be easily extended to DNNs with general piecewise linear activation functions, we focus on DNNs with ReLU activations in the rest of the paper for clarity. In addition to the piecewise linear property, ReLU units are cheap to compute, as is their gradient, and are widely applicable to many different tasks while achieving good performance (He et al., 2016a; Zagoruyko & Komodakis, 2016). In the following, we will study how dropout improves the generalization performance of a complicated DNN by considering how it generalizes each local linear model to its nearby convex polyhedra. This is easier to analyze and acts as the inspiration for our modifications to the original dropout. We will further elaborate the understandings of dropout based on the above insights of local linear models in the next section.

## 3 THREE MODIFICATIONS TO DROPOUT LEAD TO JUMPOUT

### 3.1 MODIFICATION I: MONOTONE DROPOUT RATE FOR LOCAL SMOOTHNESS

There have been multiple explanations for how dropout improves the performance of DNNs. Firstly, dropout prevents the co-adaptation of the neurons in the network, or in other words, encourages the independence or diversity amongst the neurons. Secondly, by randomly dropping a portion of neurons during training, we effectively train a large number of smaller networks, and during test/inference, the network prediction can be treated as an ensemble of the outputs from those smaller networks, and thus enjoys the advantages of using an ensemble such as variance reduction.

Here we provide another perspective for understanding how dropout improves generalization performance by inspecting how it smooths each local linear model described in the previous section. As mentioned above, for a DNN with ReLUs, the input space is divided into convex polyhedra, and for any data point in every convex polyhedron of the final layer (a polyhedron that is not divided further into smaller regions), the DNN behaves exactly as a linear model. For large DNNs with thousands of neurons per layer, the number of convex polyhedra can be exponential in the number of neurons. Hence, there is a high chance that the training samples will be dispersedly situated amongst the different polyhedra, and every training data point is likely to be given its own distinct local linear model. Moreover, it is possible that two nearby polyhedra may correspond to arbitrarily different linear models, since they are the results of consecutively multiplying a series of weight matrices $W_m W_{m-1}^x \dots W_1^x$ of different $x$ (as shown in Eqn. (2)), where each weight matrix $W_j^x$ is $W_j$ with some rows setting to be all-zero according to the activation pattern $a_j(x)$ of a specific data point $x$. If the activation patterns of two polyhedra differ on some critical rows of the weight matrices, the resulting linear models may differ a lot. Therefore, it is possible that the linear model of one polyhedron can only work for one or a few training data points strictly within the polyhedron, and

may fail when applied to any nearby test data point (i.e., a lack of smoothness). This might make DNN fragile and perform unstably on new data, and thus weaken its generalization ability.

Given the problems of dropout mentioned in Section 1.1, we propose to sample a dropout rate from a truncated half-normal distribution (to get a positive value), which is the positive part of an ordinary Gaussian distribution with mean zero. In particular, we firstly sample $p \sim \mathcal{N}(0, \sigma)$ from a Gaussian distribution, and then take the absolute value $|p|$ as the dropout rate. We further truncate $|p|$ so that $|p| \in [p_{\min}, p_{\max}]$, where $0 \leq p_{\min} < p_{\max} \leq 1$. These determine the lower and upper limits of the dropout rate and are used to ensure that the sampled probability does not get either too small, which makes jumpout ineffective, or too large, which may yield poor performance. Overall, this achieves a monotone decreasing probability of a given dropout rate as mentioned above. Other distributions (such as a Beta distribution) could also be used for this purpose, but we leave that to future work.

We utilize the standard deviation $\sigma$ as the hyper-parameter to control the amount of generalization enforcement. By using the above method, smaller dropout rates are sampled with higher probabilities so that a training sample will be more likely to contribute to the linear models of closer polyhedra. Therefore, such a Gaussian-based dropout rate distribution encourages the smoothness of the generalization performance of each local linear model, i.e., it will still perform well on points in closer polyhedra, but its effectiveness will diminish for a point farther away from the polyhedron it belongs to.

### 3.2 MODIFICATION II: DROPOUT RATE ADAPTED TO THE NUMBER OF ACTIVATED NEURONS

The dropout rate for each layer is a hyper-parameter, and as stated above, it controls a form of smoothness amongst nearby local linear models. Ideally, the dropout rates of different layers should be tuned separately to improve network performance. In practice, it is computationally expensive or infeasible to tune so many hyper-parameters. One widely adopted approach is therefore to set the same drop rate for all layers and to tune one global dropout rate.

Using a single global dropout rate is suboptimal because the proportion of active neurons (i.e., neurons with positive values) of each layer at each training stage and for each sample can be dramatically different (see Fig. 1). When applying the same dropout rate to different layers, different fractions of active neurons get deactivated, so the effective dropout rate applied to the active neurons varies significantly. Suppose the fraction of active neurons in layer $j$ is $q_j^+ = (\sum_{i=1:d} \mathbb{1}_{h_j[i]>0})/|h_j|$. Since dropping the inactive neurons has no effects (neurons with values $\leq 0$ have already been set to 0 by ReLU), the effective dropout rate of every layer is $p_j q_j^+$, where $p_j$ is the dropout rate of layer $j$. Thus, to better control the behavior of

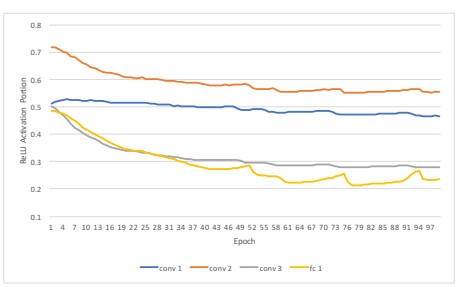

Figure 1: Portion of activate neurons on different layers throughout the training process. The network is "CIFAR10(s)" (see Sec. 4).

dropout for different layers and across various training stages, we normalize the dropout rate by $q_j^+$ and use an actual dropout rate of $p_j' = p_j/q_j^+$. By doing so, the hamming distance between the changed activation pattern and the original pattern is more consistent, and we can more precisely achieve the desirable level of smoothing encouragement by tuning the dropout rate as a single hyper-parameter.

### 3.3 MODIFICATION III: RESCALE OUTPUTS TO WORK WITH BATCH NORMALIZATION

In standard dropout, if the dropout rate is $p$, we scale the neurons by $1/p$ during training and keeps the neuron values unchanged during the test/inference phase. The scaling factor $1/p$ keeps the mean of the neurons the same between training and test; this constitutes a primary reason for the incompatibility between dropout and batch normalization (BN) (Ioffe & Szegedy, 2015). Specifically, though the mean of neurons is consistent, the variance can be dramatically different between the training and test phases, in which case the DNN might have unpredictable behavior as the BN layers cannot adapt to the change of variance from training to test condition.

We consider one possible setting of combining dropout layers with BN layers where one linear computational layer (e.g., a fully-connected or a convolutional layer without activation function) is followed by a BN layer, then a ReLU activation layer, and then followed by a dropout layer. For layer $j$, without

loss of generality, we may treat the value of a neuron $i$ after ReLU, i.e., $\bar{h}_j[i]$ as a random variable with $q_j^+$ probability of being 1 and $1 - q_j^+$ probability of being 0. If dropout is not applied, $\bar{h}_j[i]$ then gets multiplied by certain entry in the weight matrix $W_{j+1}[i', i]$, and contributes to the value of the $i'$ neuron of layer $j+1$. Since we consider any index $i$ and $i'$, we rename the following terms for simplicity: $x_j := \bar{h}_j$, $w_j := W_{j+1}[i', i]$, $y_j := h_{j+1}[i']$. As neuron $i'$ of layer $j+1$ (before ReLU) then gets fed into a BN layer, we will focus on the change of mean and variance as we add the dropout layer.

Suppose we apply a dropout rate of $p_j$, then

$$E[y_j] = E[w_j](1 - p_j)q_j^+ \tag{3}$$

$$Var[y_j] = E[y_j^2] - E[y_j]^2 = (1 - p_j)E[(w_j x_j)^2] - (E[w_j](1 - p_j)q_j^+)^2 \tag{4}$$

Hence, dropout changes both the scales of the mean and variance of neurons during training. Since the following BN's parameters are trained based on the scaled mean and variance, which however are not scaled by dropout during test/inference (because dropout is not used during testing), the trained BN is not consistent with the test phase. An easy fix of the inconsistency is to rescale the output $y_j$ to counteract dropout's on the scales of mean and variance. In order to recover the original scale of the mean, we should rescale the dropped neurons by $(1 - p_j)^{-1}$. However, the rescaling factor should be $(1 - p_j)^{-0.5}$ instead for recovering the scale of the variance if $E(y_j)$ is small and thus the second term of the variance can be ignored.

Ideally, we can also take into account the value of $E[w_j]$, and scale the un-dropped nuerons by

$$\sqrt{\frac{E[(w_j x_j)^2] - (E[w_j]q_j^+)^2}{(1 - p_j)E[(w_j x_j)^2] - (E[w_j](1 - p_j)q_j^+)^2}}. \tag{5}$$

However, computing information about $w_j$, which is the weight matrix of the following layer, requires additional computation and memory cost. In addition, such a scaling factor is only correct for the variance of $y_j$. To make the mean consistent, we should instead use $(1 - p_j)^{-1}$ (the original dropout scaling factor). No simple scaling method can resolve the shift in both mean and variance, as the mean rescaling $(1 - p_j)^{-1}$ does not solve the variance shift.

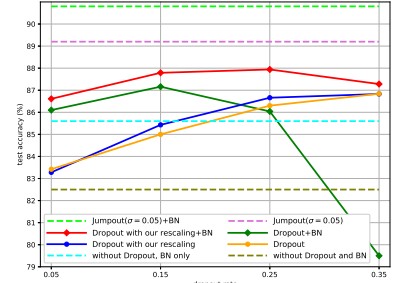

Figure 2: Comparison of the original dropout, dropout with our rescaling and jumpout, on their performance (after 150 training epochs) when used with or without bactch normalization (BN) in "CIFAR10(s)" network (see Sec. 4). Jumpout will be formally introduced in Sec. 3.4.

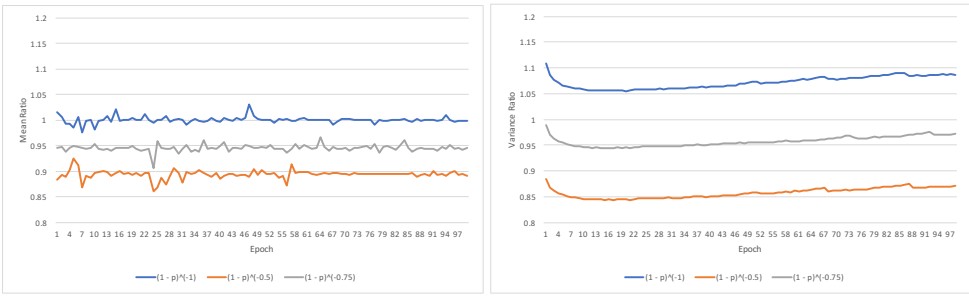

Figure 3: Comparison of mean/variance drift when using $(1 - p)^{-1}$, $(1 - p)^{-0.5}$ and $(1 - p)^{-0.75}$ as the dropout rescaling factor applied to $y$, when $p = 0.2$. The network is "CIFAR10(s)" (see Sec. 4). The left plot shows the empirical mean of $y$ with dropout divided by the case without dropout (averaged over all layers), and the second plot shows the similar ratio for the variance. Ideally, both ratios should be close to 1. As shown in the plots, $(1 - p)^{-0.75}$ gives nice trade-offs between the mean and variance rescaling.

When the mean $E(y_j)$ is large in magnitude, so that the second term in the variance is comparable with the first term, in which case the variance is small, we should use the rescaling factor close to $(1-p_j)^{-1}$, which makes the mean exactly unchanged for training and test. In contrast, when the mean $E(y_j)$ is small in magnitude and close to 0, the second term in the variance is ignorable, and we should use $(1 - p_j)^{-0.5}$ as the rescaling factor, to make the variance unchanged. In practice, it is not efficient to compute $E(y_j)$ during training, so we propose to use a trade-off point $(1 - p_j)^{-0.75}$ between $(1-p_j)^{-1}$ and $(1 - p_j)^{-0.5}$. In Figure 3, we show that $(1-p_j)^{-0.75}$ makes both the mean and variance sufficiently consistent for the cases of using dropout and not using dropout. In Figure 2, we compare the performance of the original dropout and dropout using our rescaling factor $(1-p_j)^{-0.75}$, when they

---

**Algorithm 1:** Jumpout Layer

**input** : $h_j, \sigma, p_{max}, p_{min}$

1 $q_j^+ := (\sum_{i=1:d_j} \mathbb{1}_{h_j[i]>0})/|h_j|$ ;                    // Compute the fraction of activated neurons

2 $p \sim \mathcal{N}(0,\sigma), p_j := \min(p_{min} + |p|, p_{max})$;                    // Sample a Gaussian dropout rate

3 $p_j' := p_j/q_j^+$;            // Normalize the dropout rate according to the fraction of activated neurons

4 Randomly generate a 0/1 mask $z_j$ for $h_j$, with probability $p_j'$ to be 0;       // Sample the dropout mask

5 $s_j := (1 - p')^{-0.75}$;                    // Compute the rescaling factor

6 $h_j' := s_j * \text{diag}(z_j)h_j$;                    // Rescale the outputs

7 **return** $h_j'$

---

**Algorithm 1:** Jumpout layer for DNN with ReLU.

are used with and without BN in a convolutional networks. It shows that using dropout with BN can potentially improve the performance, and larger dropout might result in more improvement. However, using the original dropout with BN leads to a significant decrease in the accuracy once increasing the dropout rate over 0.15. In contrast, the performance of dropout using our rescaling keeps improving with increasing dropout rate (until reaching 0.25), and is the best among the four configurations.

### 3.4 JUMPOUT LAYER

We combine the three modifications specifically designed to overcome the drawbacks of the original dropout in our proposed improved dropout, which we call "Jumpout" as shown in Alg. 1. Similar to the original dropout, jumpout essentially generates a 0/1 mask for the input neurons, and randomly drop a portion of the neurons based on the mask.

Summarizing the novelty of jumpout, instead of using a fixed dropout rate as in the original dropout, jumpout samples from a monotone decreasing distribution as mentioned above to get a random dropout rate. Also, jumpout normalizes the dropout rate adaptively based on the number of active neurons, which enforces consistent regularization and generalization effects on different layers, across different training stages, and on different samples. Finally, jumpout further scales the outputs by $(1 - p)^{-0.75}$, as opposed to $(1 - p)^{-1}$ during training, in order to trade-off the mean and variance shifts and synergize well with batchnorm operations.

Jumpout requires one main hyper-parameter $\sigma$ to control the standard deviation of the half-normal distribution, and two auxiliary truncation hyperparameters $(p_{min}, p_{max})$. Though $(p_{min}, p_{max})$ can also be tunned, they serve to bound the samples from the half-normal distribution; in practice, we set $p_{min} = 0.01$ and $p_{max} = 0.6$, which work consistently well over all datasets and models we tried. Hence, jumpout has three hyperparameters, although we only tuned $\sigma$ and achieved good performance, as can be seen below.

Also, note that here we consider the input $h_j$ to be the features of layer $j$ corresponding to one data point. For a mini-batch of data points, we can either estimate $q_j^+$ separately for each single data point in the mini-batch or apply the average $q_j^+$ over data points as the estimate for the mini-batch. In practice, we utilize the latter option as we find that it gives comparable performance to the first while using less computation and memory.

Jumpout has almost the same memory cost as the original dropout, which is the additional 0/1 drop mask. For computation, jumpout requires counting the number of active neurons, which is insignificant compared to the other layers of a deep model, and sampling from the distribution, which is also insignificant compared to the other computation in DNN training.

## 4 EXPERIMENTS

In this section, we apply dropout and jumpout to different popular DNN architectures and compare their performance on six benchmark datasets at different scales. In particular, these DNN architectures include a small CNN with four convolutional layers[2] applied to CIFAR10 (Krizhevsky & Hinton, 2009), WideResNet-28-10 (Zagoruyko & Komodakis, 2016) applied to CIFAR10 and CIFAR100 (Krizhevsky & Hinton, 2009), "pre-activation" version of ResNet-20 (He et al., 2016b) applied to Fashion-MNIST ("Fashion" in all tables) (Xiao et al., 2017), WideResNet-16-8 applied

---

[2]The "v3" network from `https://github.com/jseppanen/cifar_lasagne`.

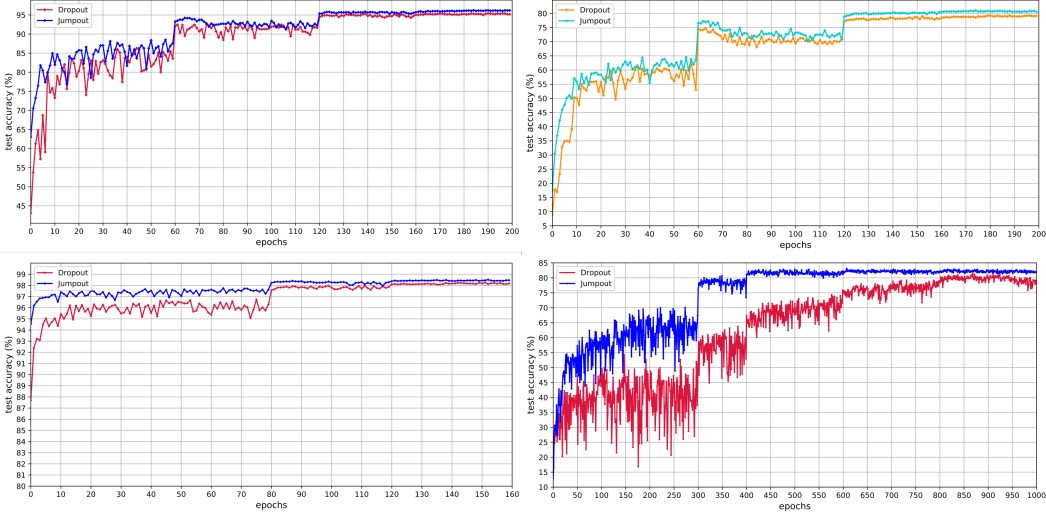

Figure 4: **Top Left**: WideResNet-28-10+Dropout and WideResNet-28-10+Jumpout on CIFAR10; **Top Right**: WideResNet-28-10+Dropout and WideResNet-28-10+Jumpout on CIFAR100; **Bottom Left**: WideResNet-16-8+Dropout and WideResNet-16-8+Jumpout on SVHN; **Bottom Right**: WideResNet-16-8+Dropout and WideResNet-16-8+Jumpout on STL10.

to SVHN (Netzer et al., 2011) and STL10 (Coates et al., 2011), and ResNet-18 (He et al., 2016a) applied to ImageNet (Deng et al., 2009; Russakovsky et al., 2015). The information about the all the datasets can be found in Table 4 at Appendix.

For all the experiments about CIFAR and Fashion-MNIST, we follow the standard settings, data preprocessing/augmentation, and hyperparameters used in an existing GitHub repository [3]. On ImageNet, we starts from a pre-trained ResNet18 model[4], and train two copies of it with dropout and jumpout respectively for the same number of epochs. The reason for not starting from random initialized model weights is that training DNNs on ImageNet usually

Table 1: Ablation study (test accuracy in %) of all the possible combinations of the three modifications (I, II and III) in jumpout, "CIFAR10(s)" refers to the small CNN applied to CIFAR10.

| Dataset | CIFAR10(s) | CIFAR10 | CIFAR100 | STL10 |
|---|---|---|---|---|
| Dropout | 86.50 | 95.23 | 79.41 | 81.37 |
| Dropout+I | 87.56 | 96.06 | 79.89 | 81.76 |
| Dropout+I+II | 90.06 | 96.35 | 81.70 | 82.09 |
| Dropout+II | 87.14 | 95.67 | 80.24 | 81.94 |
| Dropout+II+III | 89.64 | 96.74 | 80.61 | 82.22 |
| Dropout+III | 87.70 | 96.20 | 80.59 | 81.69 |
| Dropout+III+I | 87.36 | 96.45 | 81.20 | 82.18 |
| Jumpout | 90.24 | 96.82 | 82.48 | 84.02 |

does not have overfitting problem if one follows the standard data augmentation methods used to train most modern models, but both dropout and jumpout are most effective in the case of overfitting. Therefore, we choose to start from the pre-trained model, on which training accuracy is relatively high (but still not overfit and very close to the test accuracy)[5].

Table 2: Final performance (test accuracy in %) of different DNNs using dropout and jumpout.

| Dataset | CIFAR10(s) | CIFAR10 | CIFAR100 | Fashion | STL10 | SVHN | ImageNet |
|---|---|---|---|---|---|---|---|
| Dropout | 86.50 | 95.23 | 79.41 | 96.09 | 81.37 | 98.22 | 71.15 |
| Jumpout | 90.24 | 96.82 | 82.48 | 97.17 | 84.02 | 98.51 | 71.48 |

We summarize the experimental results in Table 2 which shows that jumpout consistently outperforms dropout on all datasets and all the DNNs we tested. Moreover, for Fashion-MNIST and CIFAR10 on which the test accuracy is already $> 95\%$, jumpout can still bring appreciable improvements. In addition, on CIFAR100 and ImageNet (on which a great number of DNNs and training methods are heavily tunned), jumpout achieves the improvement that can only be obtained by significantly

---

[3]https://github.com/hysts/pytorch_image_classification

[4]https://gluon-cv.mxnet.io/api/model_zoo.html#gluoncv.model_zoo.resnet18_v1b

[5]In fact, ImageNet is less of an appropriate benchmark to test the performance of dropout, and almost all the modern DNNs for ImageNet do not use dropout, because the training accuracy is always close to the test accuracy during the training process, and there is no overfitting problem needed to be tackled by dropout. We include ImageNet in experiments because of its large size compared to the other datasets used.

increasing the model size in the past. These verify the effectiveness of jumpout and its advantage comparing to the original dropout.

In addition, we conduct a thorough ablation study of all the possible combinations of the three proposed modifications, with results reported in Table 1. It further verifies the effectiveness of each modification: 1) each modification improves the vanilla dropout; 2) adding any modification to another brings further improvements; and 3) applying the three modifications together (i.e., jumpout) achieves the best performance.

We also provide the learning curves and convergence plots of dropout and jumpout equipped DNNs during training in Figure 4. In all the figures, "Jumpout" applies adaptive dropout rate per mini-batch. Jumpout exhibits substantial advantages over dropout in early learning stages, and reaches a reasonably good accuracy much faster. In the future, it may be possible to find a better learning rate schedule method specifically for jumpout, so it can reach the final performance earlier than dropout.

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

# 5 APPENDIX

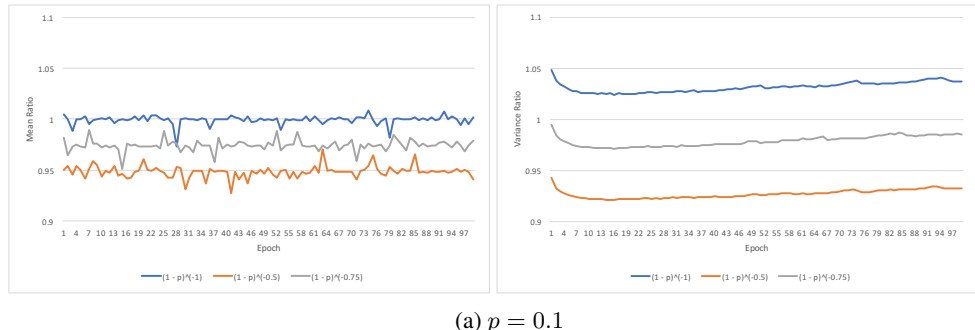

(a) $p = 0.1$

Figure 5: Comparison of mean/variance drift when using $(1 - p)^{-1}$, $(1 - p)^{-0.5}$ and $(1 - p)^{-0.75}$ as the dropout rescaling factor applied to $y$, when $p = 0.1$. The network is "CIFAR10(s)" (see Sec. 4). The left plot shows the empirical mean of $y$ with dropout divided by the case without dropout (averaged over all layers), and the second plot shows the similar ratio for the variance. Ideally, both ratios should be close to 1. As shown in the plots, $(1 - p)^{-0.75}$ gives nice trade-offs between the mean and variance rescaling.

Table 3: Comparison (test accuracy in %) to concrete dropout (a variational dropout method) Gal et al. (2017).

| Dataset | CIFAR10 | CIFAR100 | STL10 |
|---|---|---|---|
| Dropout | 95.23 | 79.41 | 81.37 |
| Concrete Dropout Gal et al. (2017) | 95.29 | 79.06 | 81.87 |
| Jumpout | 96.82 | 82.48 | 84.02 |

Table 4: Details regarding the datasets.

| Dataset | CIFAR10 | CIFAR100 | Fashion | STL10 | SVHN | ImageNet |
|---|---|---|---|---|---|---|
| #Training | 50000 | 50000 | 60000 | 5000 | 604388 | 1281166 |
| #Test | 10000 | 10000 | 10000 | 8000 | 26032 | 50000 |
| #Class | 10 | 100 | 10 | 10 | 10 | 1000 |
| #Feature | $3 \times 32 \times 32$ | $3 \times 32 \times 32$ | $28 \times 28$ | $3 \times 96 \times 96$ | $3 \times 32 \times 32$ | $3 \times 224 \times 224$ |

