# OpenReview forum: "Jumpout: Improved Dropout for Deep Neural Networks with Rectified Linear Units"
_ICLR.cc/2019/Conference_

### Official Review · AnonReviewer2 · 2018-10-14
**Unconvincing experimental results and theoretical arguments**

**Rating:** 4
**Confidence:** 5

**Review:**

The paper proposes yet another variant of the celebrated Dropout algorithm. Specifically, the proposed method attempts to address the obvious drawbacks of Dropout: (i) the need to heuristically select the Dropout rate; and (ii) the universality of this selection across a layer.

As the authors have admitted in the paper (Sec. 1.2), there is a variety of methods already addressing the same problem. They argue that contrary to some of these methods "jumpout does not rely on additional trained models: it adjusts the dropout rate solely based on the ReLU activation patterns. Moreover, jumpout introduces negligible computation and memory overhead relative to the original dropout methods, and can be easily incorporated into existing model architectures."

However, this is argument is certainly untrue and rather misleading. The works of Kingma et al. (2015) and Molchanov et al. (2017), that the authors cite, does not introduce additional trained models. In addition, there is additional related work that the authors do not cite, but ought to:

[1] Yarin Gal, Jiri Hron, Alex Kendall, "Concrete Dropout," Proc. NIPS 2017.
[2] Yingzhen Li, Yarin Gal, "Dropout Inference in Bayesian Neural Networks with Alpha-divergences," Proc ICML 2017.
[3] Harris Partaourides, Sotirios Chatzis, “Deep Network Regularization via Bayesian Inference of Synaptic Connectivity,” J. Kim et al. (Eds.): PAKDD 2017, Part I, LNAI 10234, pp. 30–41, 2017.

These methods also address a similar problem, without introducing extra networks or imposing extra costs art inference time. Thus, citing them, as well as COMPARING to them, is a necessity for this paper to be convincing.

These crucial shortcoming aside, there are various theoretical claims in this paper that are not sufficiently substantiated. To begin with, the arguments used in the last paragraph of page 4 seem at least speculative; then,  the authors proceed to propose a solution to the alleged problem in the beginning of page 5. They suggest sampling from a truncated Gaussian, but they do not elaborate on why this selection solves the problem; they limit themselves to noting that other selections, such as the Beta distribution, may also be considered in the future. We must also underline that [3] have suggested exactly that; sampling from a Beta.

Finally, the last two modifications the authors propose seem reasonable, yet they are extremely heuristic. No one knows (which can be guaranteed through theoretical proofs or solid experimental evidence) that without these the method would not work. In addition, previous papers, e.g. [1-3] achieve similar goals in a principled fashion (ie by inferring proper posterior densities); without experimental comparisons, nobody knows which paradigm is best to adopt.

---

> ### Author Response · Authors · 2018-11-27
> **Jumpout is not a variational dropout approach; Added comparison to concrete dropout**
>
> Thanks for your comments! We added the ablation study to demonstrate the effectiveness of every individual modification. We further emphasize that jumpout is not a variational dropout approach, and can scale to very large networks. We also added a comparison with concrete dropout[1] as suggested (Table 3 in Appendix).
>
> Q1: Overview of the paper: "Specifically, the proposed method attempts to address the obvious drawbacks of Dropout: (i) the need to heuristically select the Dropout rate; and (ii) the universality of this selection across a layer."
>
> R1: "(i) the need to heuristically select the Dropout rate" is merely one observation of the paper and 1/3 of the drawbacks we aim to address, and we never attempt to address "(ii) the universality of this selection across a layer", i.e., for all nodes on a layer, jumpout applies the same drop rate. The primary purpose of jumpout is to improve the original dropout performance without introducing extra computational costs. The truncated Gaussian distribution aims to improve the dropout performance based on the linear model geometry of ReLU networks. The change of dropout rate based on ReLU pattern tries to address the dropout rate selection problem. The change of rescaling factor resolves the disharmony between dropout and batchnorm, so for a network with both kinds of layers, the performance gets boosted.
>
> Q2: Comparison to [1],[2] and [3].
>
> R2: We note that jumpout is NOT a variational approach of dropout, which does not require Bayesian training or inference. Jumpout does not introduce extra inference cost, and it also has similar training costs as the original dropout. Jumpout can therefore work on modern networks with deep and wide structures, whereas the variational approaches [2] and [3] do not scale to the networks we include in the paper. For [1], we add comparison experiments and show that jumpout significantly outperforms [1].
> [1],[2] and [3] tries to address the problem similar to our observation 2, namely, selection of the dropout rate. Jumpout has 2 other major changes: we choose to impose a truncated Gaussian distribution on the dropout rate based on the linear model geometry of the ReLU network, and we change the rescaling factor to account for the disharmony between dropout and batchnorm.
>
> Q3: They suggest sampling from a truncated Gaussian, but they do not elaborate on why this selection solves the problem.
>
> R3:The truncated Gaussian distribution is a natural choice based on the intuition based on the linear model geometry of ReLU networks. Again, jumpout is not a variational approach, and the truncated Gaussian distribution is not aimed to solve the dropout rate selection problem. The truncated Gaussian is applied because the original dropout has the uniform preference for both nearby and faraway linear models, while in principle, close linear models should be preferred. Also, for [3], the beta distribution is selected with no clear reason at all.
>
> Q4: No one knows without these the method would not work.
>
> R4: We add thorough ablation studies on all combinations of the 3 modifications to show that 1) they all have positive impacts on the performance, and 2) 3 modifications can work together to get the best performance.

---

### Official Review · AnonReviewer1 · 2018-11-03
**not well explained and not rigorously tested**

**Rating:** 4
**Confidence:** 3

**Review:**

This paper proposes jumpout, which is a 3 step modification based on dropoout
that is designed to work better with batch normalization. Unfortunately, I did not understand the arguments on locally linear regions and ReLu and its relationship with the monotone dropout scheme,
or why the half Gaussian is chosen.

Still, jump out the procedure is fairly clear in Algorithm 1, and the results seems good.
However, I could not make out much of why each step is done, and could not find empirical tests of the value of each step.

I think the paper needs more work. All the proposals seem very heuristic, and it is important to test their separate effects. It should be easy to perform a ablation analysis since the 3 proposed steps are pretty independent and can be tested separately. Since two of these have to do with modifying the dropout rate, it would be important to compare with carefully cross-validated dropout rates, which I also do not see.

---

> ### Author Response · Authors · 2018-11-27
> **Ablation Studies added; Theoretical supports**
>
> Thanks for your comments! We added the ablation study you suggested, and briefly explained the locally linear region in the following. The three modifications are based on theoretical analysis and are not heuristic.
>
> Q1: Unfortunately, I did not understand the arguments on locally linear regions and ReLu and its relationship with the monotone dropout scheme, or why the half Gaussian is chosen.
>
> R1: Intuitively, we show that a ReLU DNN equals to a set of linear models defined on a set of respective convex polyhedra in the input space. Each linear model is only applied to the data point within the respective polyhedron. The monotone dropout rate encourages the local smoothness of the generalization of each linear model, by training the linear model also on data points located at other nearby polyhedra with higher probability. Sampling from the half Gaussian ensures that the data points from closer polyhedra have a higher probability to be used to train the linear model.
>
> Q2: However, I could not make out much of why each step is done, and could not find empirical tests of the value of each step...it is important to test their separate effects...It should be easy to perform an ablation analysis...
>
> R2: In the updated draft (Table 1), we provided a thorough ablation study on multiple datasets. It compares the performance of all the 7 different combinations of the three modifications. This will provide a complete answer to your question.
>
> Q3: All the proposals seem very heuristic.
>
> R3: This is not true. Modification 1 and 2 are theoretically supported by the rigorous analysis of ReLU DNNs in Section 2, while modification 3 is derived from the given analysis of mean/variance drift in Section 3.3 (it aims to balance the reduced mean drift and variance drift).

---

### Official Review · AnonReviewer3 · 2018-11-05
**needs further empirical evidence**

**Rating:** 5
**Confidence:** 4

**Review:**

Authors propose three modifications to dropout, specifically in context of dropout applied to deep networks utilizing the ReLU non-linearity.  The three modifications seem independently motivated and aim to overcome separate potential shortcomings of the current dropout approach.  These three modifications are combined into a new approach termed Jumpout.

Overall I find this to be a weak paper requiring further work, for the following main reasons:

* The proposed modifications are intuitively motivated and then empirically supported.  However, I find the intuitive reasoning unclear and have to lean much more on empirical evidence.  For instance, the motivation for modification 2 “dropout rate adapted to number of active neurons”, is that in case ReLU causes a large number of neurons to ‘shut down’ then the dropout rate in that layer should be reduced (or increased, depending on how it is defined) causing fewer neurons to further dropout.  However, if preventing co-adaptation is a reason to dropout neurons then the issue of conditional correlation (or co-activation given related inputs) will remain regardless of number of active neurons in a layer, thus changing the dropout rate as a function of ReLU activation is not fully justified.  Similarly, modification 3 “rescale outputs to work with batch normalization” proposes exponentiation by -0.75 with weak justification as a compromise.

* I find the empirical evidence and support for the three modifications lacking in detail.  The authors provide results of the combined Jumpout technique on a number of tasks, but do not demonstrate effectiveness and contribution of individual modifications on error rates on the tasks they evaluated.

* I also find the baseline systems to be on the weaker side (e.g. on CIFAR100 many systems now have higher than 82% accuracy with best being over 84, on STL-10 many systems now are well above 85%).

---

> ### Author Response · Authors · 2018-11-27
> **Ablation Studies added; Theoretical supports, not heuristics (2)**
>
> Q4: I find the empirical evidence and support for the three modifications lacking in detail.  The authors provide results of the combined Jumpout technique on a number of tasks, but do not demonstrate the effectiveness and contribution of individual modifications on error rates on the tasks they evaluated.
>
> R4: We provided a thorough ablation study on multiple datasets in the updated draft (Table 1). The ablation study compares the performance of all the 7 different combinations of the three modifications. It shows that 1) each modification brings improvement to the vanilla dropout; 2) adding any modification to another brings further improvements; and 3) applying the three modifications together achieves the best performance.
>
> Q5: I also find the baseline systems to be on the weaker side (e.g. on CIFAR100 many systems now have higher than 82% accuracy with best being over 84, on STL-10 many systems now are well above 85%).
>
> R5: 1) It is not fair to justify a dropout technique by comparing its performance on two different systems; 2) The purpose of this paper is not to pursue SOTA performance on CIFAR100 and STL10 by combining several complicated tricks or employing an extremely large and costly model. Instead, our goal is to provide an easy-to-implement, efficient and effective dropout technique. This has been verified by the experimental results, i.e., jumpout always brings promising improvements (~2% on CIFAR100 and >2.3% on STL10) on the same model (WideResNet and ResNet) with negligible extra computation; 3) Achieving SOTA performance is usually much more expensive, either due to the extremely large size of model or complicated data augmentation/regularization; 4) dropout/jumpout improves generalization by preventing overfitting, which usually happens when training relatively small neural networks on small data (CIFAR100 and STL10), but not for severely over-parameterized models achieving SOTA performance (recent theoretical papers proved that over-parameterized model is harder to overfit).

---

> ### Author Response · Authors · 2018-11-27
> **Ablation Studies added; Theoretical supports, not heuristics (1)**
>
> Thanks for your comments! We add a thorough ablation study as you suggested, but do not agree with other points.
>
> Q1: However, I find the intuitive reasoning unclear and have to lean much more on empirical evidence.
>
> R1: Modification 1 and 2 are theoretically supported by the rigorous analysis of ReLU DNNs in Section 2, i.e., a ReLU DNN equals to a set of local linear models defined on a set of respective convex polyhedra in the input space, each containing a few data points. Modification 1 improves the local smoothness of the generalization of each linear model (associated with a specific polyhedron), by training it also on data points located at other nearby polyhedra with higher probability. Modification 2 ensures the homogeneity of the local smoothness, i.e., it generalizes each linear model to the nearby polyhedra of equal distance to the original one (measured by the number of different activation patterns) with the same probability. Modification 3 aims to reduce and balance the mean drift and variance drift when applying dropout together with batch normalization.
>
> Q2: For instance, the motivation for modification 2...However, if preventing co-adaptation is a reason to dropout neurons then the issue of conditional correlation (or co-activation given related inputs) will remain regardless of the number of active neurons in a layer, thus changing the dropout rate as a function of ReLU activation is not fully justified.
>
> R2: It is wrong to entirely block co-adaptation. Dropout aims to weaken co-adaptation but not to entirely remove it, since exploring the correlation between hidden nodes is an important part of optimizing the model weights (considering backpropagation for example). Comparing to dropout, jumpout allows slightly more co-adaptation, but the amount is extremely small and negligible. Because the adaptive dropout rate is a single number applied to hundreds of thousands of hidden nodes in a layer and a mini-batch. Considering how much a single number can describe the correlation among hundreds of thousands of variables: its influence is negligible.
>
> In addition, it is worth noting that fixing dropout rate can be catastrophic during training. As shown in Figure 1, since existing training methods do not have any control on the ratio of activated neurons per layer, it is very possible that some layers have many activated nodes while some have very few. For the former, a relatively large dropout rate is required to avoid overfitting. However, applying the same large dropout rate to the latter will almost cut the information flow sent from input to deeper layers. In this case, the output will almost independent to the input, which is catastrophic.
>
> Q3: Similarly, modification 3 “rescale outputs to work with batch normalization” proposes exponentiation by -0.75 with weak justification as a compromise.
>
> R3: Modification 3 is theoretically derived from the given analysis of mean/variance drift in Section 3.3. In order to balance the reduced mean drift and variance drift, the power in the rescaling factor should be between $-0.5$ and $-1.0$. Without any extra information about the weight matrices of the following layers, -0.75 provides a good trade-off between reducing the mean drift and the variance drift (as shown in Figure 3), and also shows promising and consistent performance boost in our experiments. So modification 3 does not rely on any "weak justification".

---

### Author Response · Authors · 2018-11-27
**Summary of Updates and Responses**

We appreciate all the reviewers for their comments and suggestions! As suggested by the reviewers, in the updated draft (Table 1), we added a thorough ablation study of all the possible combinations of the three modifications proposed in this paper, and show the effectiveness of each of them on four datasets.

We also emphasized in our response that the three modifications are based on rigorous analysis and new insights to ReLU networks (most in Section 1.1 and Section 2, which might be ignored but are important) rather than sheer heuristics or empirical evidence only.

In addition, we are not proposing a variational dropout method. Instead, we are modifying the vanilla dropout to make it consistent the new analysis. Jumpout requires the same cost as the vanilla dropout for both training and test, has a very simple implementation, and improves the performance consistently and dramatically.

---

### Meta-Review · Area_Chair1 · 2018-12-13

**Confidence:** 4
**Recommendation:** Reject

**Metareview:**

The paper introduces a new variant of the Dropout method. The reviewers agree that the procedure is clear. However, motivations behind the method are heuristic, and have to lean much on empirical evidence. A strong motivation behind the procedure is lacking, and the motivation behind the method is unclear. Furthermore, the empirical evidence is lacking in detail and could use better comparisons with existing literature.